

# Investigating the putative unforeseen link between football fervour and colorectal cancer screening in Denmark

Alonzo Alfaro-Núñez[1,2], Stina Christensen[2] and Esther A. Jensen[2,3]

[1] Section for Geogenetics, GLOBE Institute, University of Copenhagen, Copenhagen K, Copenhagen, Denmark
[2] Department of Clinical Biochemistry, Naestved Hospital, Naestved, Naestved, Denmark
[3] The Secretariat for Colorectal Screening, Region Zealand, Naestved, Denmark

## ABSTRACT

Colorectal cancer (CRC) ranks as the third most prevalent cancer globally, often remaining asymptomatic in its early stages but posing high mortality risks in advanced tumours. Screening for CRC (sCRC) has shown to effectively reduce both incidence and mortality rates. In this study, we investigate a potential association between a decline in sCRC participation in Denmark and a major sporting event. We conducted an association cohort study encompassing all citizens aged 50 to 74, who were invited to undergo sCRC screening in Region Zealand, Denmark, spanning from 2014 to 2022. Our analysis revealed a noticeable reduction in sCRC participation specifically during the 2-week period in autumn 2022 coinciding with the participation of the Danish football team in the Football World Cup 2022 held in Qatar. To our knowledge, this is the first instance where an international sporting event has been linked to a decline in national sCRC participation, suggesting that the fervour of sports enthusiasts may divert attention away from preventive health measures. Notably, no similar reductions in sCRC participation were observed during any other sporting events throughout the entire study period (2014–2022) in Denmark.

## INTRODUCTION

Many high-income countries, including Denmark, have since 2014 implemented national and regional monitoring efforts for the early detection of colorectal cancer (CRC), as this is the 3rd most common type of cancer worldwide (*Sung et al., 2021*; *Sinicrope, 2022*; *Bretthauer et al., 2022*). CRC is often asymptomatic in the early stages, and mortality is high in patients with late-stage tumours. Studies show that screening can reduce incidence and mortality over time (*Lauby-Secretan et al., 2018*; *Njor et al., 2018*; *Breekveldt et al., 2022*; *Bretthauer et al., 2022*; *Njor et al., 2022*). Therefore, since 2014 Denmark introduced a national screening for CRC (sCRC) where all citizens in the ages between 50 to 74 were invited once between 2014 and 2017 (*Larsen et al., 2018*; *Njor et al., 2018*), and from 2018,

Corresponding author
Alonzo Alfaro-Núñez,
alonzoalfaro@gmail.com

citizens are invited every 2 years to participate by returning a stool sample, which is tested for presence of blood (*Olesen et al., 2023*).

In Region Zealand, Denmark, only 54% of the invited men and 63% of the women return a sample for testing. It has been observed, both in European countries (*Senore et al., 2019*) and at regional levels (*Kirkegaard et al., 2018*), that compliance with the screening program can be positively affected if sCRC is featured in the news with more citizens returning samples in the following weeks. However, despite being free of charge, social inequality in the screening program has been evident among both men and women with the association between education, and nonparticipation (*Larsen et al., 2017*).

Over the years, the staff responsible for the sCRC has expressed that the busyness of the recruitment for citizen participants decreased during major sporting events, despite the data showing no variation (*Olesen et al., 2023*). Looking at the data through the years from 2014 to 2022, several significant reductions in the percentage of participation have been clearly identified. Each of these reductions could be explained by technical or logistic reasons (*e.g.*, transition to a new IT system or an erroneous inclusion of some citizens by the IT system). However, a significant decline in citizen participation was experienced during 2 weeks in the late autumn of 2022 that could not be justified by any previously known or well-characterised reason. Remarkably, an observation was made that the Danish Men's Football National team participated in the Football World Cup 2022 (FWC22) in Qatar during the same exact 2-week period (Fig. 1). Therefore, we investigate the potential intersection between these two independent variables by raising the question, could a major sport event, such as FWC22, have influenced the citizens' conscious mind to translate into a reduction of participation for the sCRC in Region Zealand, Denmark?

## METHODS

### Data sources and generation

From to 2014 to 2022, all citizens between the ages of 50–74 were invited to participate in the national screening program for CRC in Denmark *via* the national Invitation and Administration Module (IAM Production v2) (*Lund et al., 2019*; *Olesen et al., 2023*). IAM records the time invitations are created. Each region determines the number of invitations to be sent on a weekly basis. Subsequently, invitations are generated every weekend, printed during the week, and sent to the citizens, who receive an invitation and sample tube in their mailbox approximately one week later. Citizens return test tubes for analysis at their convenience. If a letter is returned due to an unknown address or if the citizen asks for a new test tube, then, the citizen will receive more than one invitation per year. For Region Zealand, a total of 1,094,336 invitations for sCRC were sent to citizens in the period from 2014 to 2022. Some of the citizens received one single invitation while some others received a maximum of four invitations depending on their age. In 2022, a total of 146,915 invitations were sent. Based on the numbers of the invitations that are sent each week and the number of citizens that respond and return a sample, the IAM provide the Secretariat for Colorectal Screening in Region Zealand with an exact record of the percentage (%) of participation response each week from 2014 to 2022 and ongoing. In this study, "*% of participation*", is calculated with the following formula:

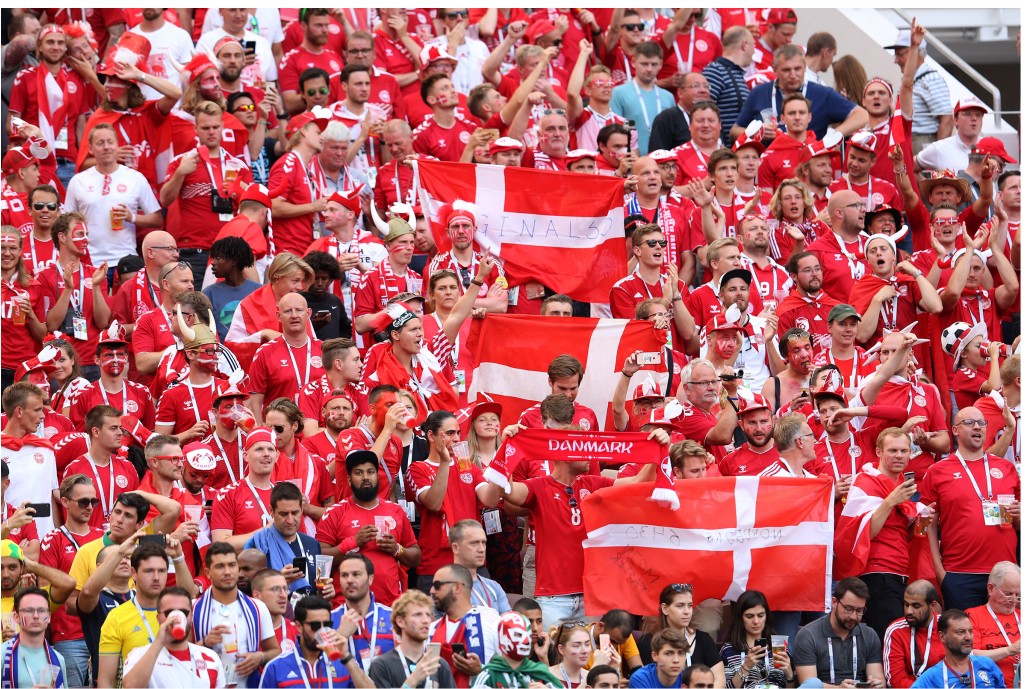

**Figure 1 Enthusiastic Danish football fans supporting their national team during the football world cup.** Photo by Marco Iacobucci. ©Shutterstock.

$$\% \ of \ participation = (number \ of \ citizens \ returning \ stool \ samples/number \ of \ invitations \ sent \ to \ the \ citizens) * 100$$

The number of citizens invited each week is not constant and varies over time due to holidays, capacity, and other logistics (*e.g.*, transitions to new IT systems, *etc.*). If an invited citizen has not returned a stool sample within a 7-week period, a reminder is sent.

It is from this day of invitation generation that the percentage of participation is determined. Thus, if an invitation was generated on Monday the 31st in October 2022 (week 44), the official letter of invitation was sent within a week to the citizen, and from there, the citizen is expected to respond in between week 46 to week 48 with a median of 31 days (Table S1). Therefore, the entire data presented in this study representing week 44 and 45, in fact correspond to the real events in weeks 47 and 48, and so on, which is the nature of our data interpretation.

According to the Danish Data Protection Agency this study does not require ethical approval (J.No. 2023-52-0016) as data was anonymized and it does not contain any personal identifiable information. All the participants in Denmark who underwent colonoscopy screening provided written informed consent.

## Data analysis

We performed a one-way ANOVA test to measure if the decline in the percentage of returned samples from citizens who received the invitation before and during the beginning of the FWC22 tournament is significantly different to the rate of participation during the same period in previous years. To determine the weekly variation within each

year and across the years, a RM (repeated measures) one-way ANOVA test was implemented to associate the total number of citizens participating in the sCRC from 2014 to 2022, and whether the time of FWC22 could have a significant reduction for the entire period of sCRC. Moreover, a one-sample t and Wilcoxon test was performed to evaluate if there was any variation in gender participation (women *vs.* men) during the 2 weeks period and all through 2022. Finally, we tested whether the group of citizens who receive the invitation during the FWC22 (generated in week 44 but received before or during the football event) have a longer median return time compared to the median for the rest of the year using a one-tailed Wilcoxon rank sum test. These tests and graphical plotting were carried out using a combination of R studio scripts version 2023.12.1+402 and GraphPad Prism software version 10.2.2.

Additionally, we included the time frames of other major sport events such as the Football World Cup (2014 and 2018), Tour de France (first three stages of the event hosted in Denmark in 2022), European Football Championship (2016 and 2021), Winter Olympics (2014, 2018, and 2022), Summer Olympics (2016, 2021), World Handball Championship (2015, 2017, 2019 and 2021) and Handball European Championships (2014, 2018, 2020 and 2022) for both women and men. Handball in particular is a very popular sport in Denmark.

## RESULTS

The one-way ANOVA test revealed that the ratio of returned samples for week 44 and 45 in 2022 were highly significantly different ($p < 0.0001$) compared with mean values during the same 2 weeks in the previous years from 2014 to 2021 (Fig. 2 and Table S2).

In 2022, weekly mean participation in the screening program was 56.7%. The percentage of returned samples dropped significantly in week 44 and 45 to 34.2% and 41.3%, respectively. The % of participation for these 2 weeks were the lowest of the year 2022 (Fig. 3A). The RM one-way ANOVA test confirmed that reduction in week 44, 2022 was highly significantly different ($p < 0.0001$) to all other weeks in the same year and in previous years (2014–2022), while week 45, 2022 appeared to be significantly different ($p < 0.05$) to most weeks for all years from 2014 to 2022 (Figs. 3B and 3C).

One-sample T-test confirmed that response rates in week 44 and week 45 were significantly lower compared with the rest of the weeks in 2022 for both men and women (Fig. 3D).

The median return time was 31 days (median absolute deviation 14.83) in 2022. Data analysis showed that the median response time for invitations generated in week 44 was significantly longer (34.2%) than the rest of the year combined ($p < 0.0001$).

No other sport events resulted in a significant reduction in participation for the sCRC (Fig. S1).

## DISCUSSION

Attendance and participation in sport activities such as football, tend to unite large audiences in the enthusiasm and commitment of watching their teams compete, and are recognised to provide positive effects for participants and spectators (*Doyle et al., 2016*).

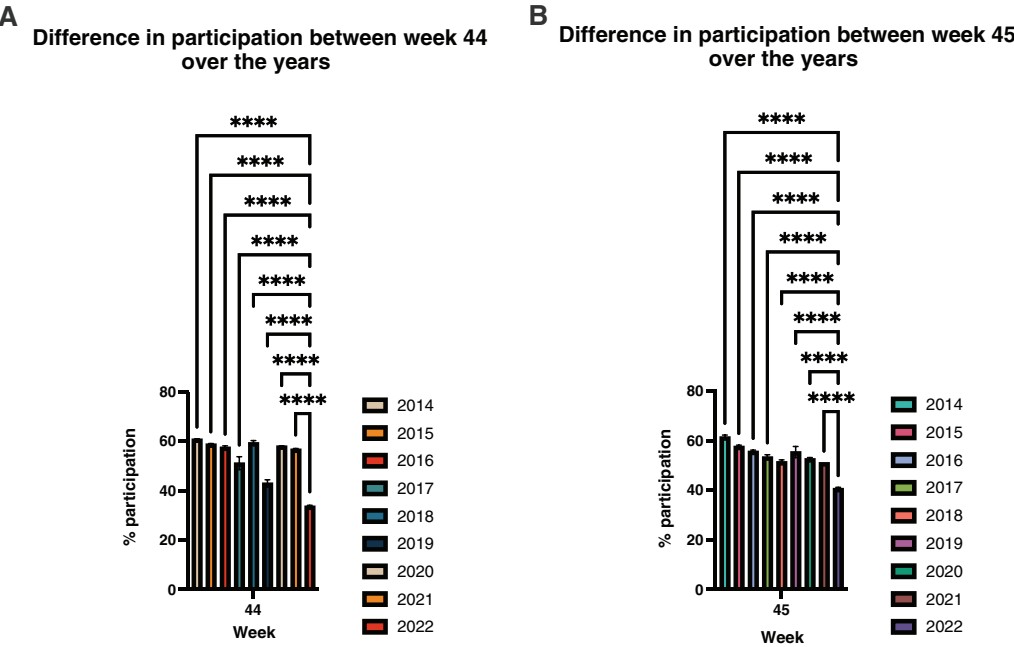

**Figure 2** There was a highly significant reduction (*p* < 0.0001) detected in the participation during week 44 (A) and week 45 (B) in 2022 compared to the same weeks in previous years since the sCRC was initiated in Denmark in 2014. Each asterisk (*) shows the level of significant difference between the mean values of the weeks compared, where four asterisks (****) are equal to *p* < 0.0001.

However, there are also well-documented negative sides associated with such massive football events, in particular related to increase in alcohol consumption, domestic violence, cardiovascular events and general negligence (*Wilbert-Lampen et al., 2008*; *Trendl, Stewart & Mullett, 2021*; *Forsdike, Hooker & Laslett, 2022*). There is also the general view that football, as one of the most popular and widely practised sports worldwide, can somehow blur the mind of the spectator (*Vallerand et al., 2008*; *Mansfield, Delia & Katz, 2020*; *Bilgiç et al., 2020*), and here we hypothesise that this blurring of mind can result in delaying other priorities, such as postponing sending samples to national screening programs.

There are many factors associated with the early detection of CRC such as genetics, microbiome composition, lifestyle including diet and fitness (*Garrett, 2019*), but we are quite certain you may have never heard of football as being one of them. After more than 2 years of pandemic with constant global shutdowns of many international sport events, the announcement that the FWC22 would not be rescheduled and would take place despite the challenges presented, was refreshing for the many millions of enthusiastic people around the world (*Dergaa et al., 2022*), including football fans in Denmark with their national team qualified for this event. In fact, Denmark was ranked ninth place together with Spain in 2022 according to the FIFA's ranking for the best teams in the world before the FWC22 in Qatar started. As such, expectations were high, and the team was domestically forecasted to go far in the tournament with an enormous national hype. Denmark is a small country with a population size at around six million, small enough for a football team to be well known and connected to the national consciousness. Football enthusiasts
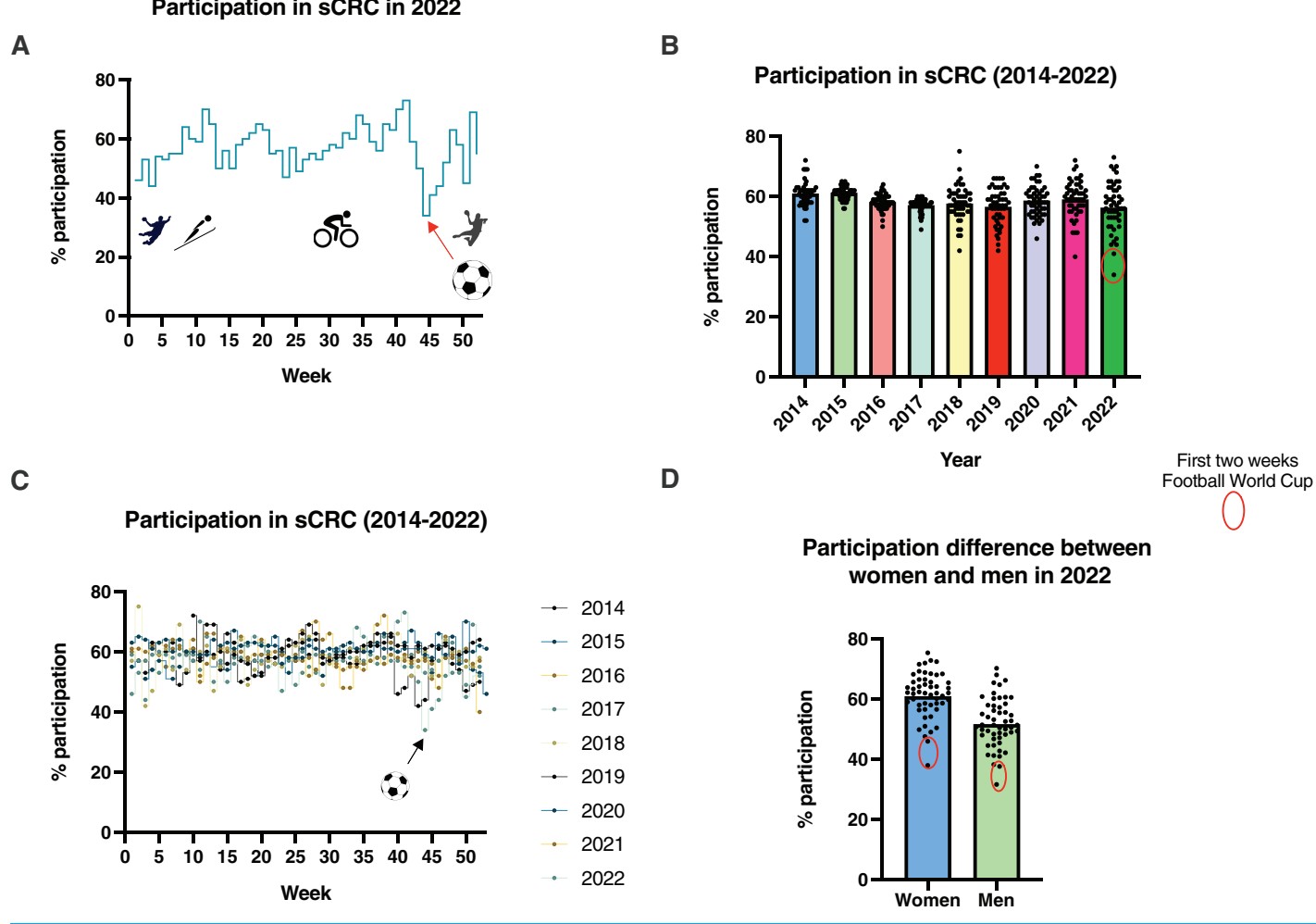

**Figure 3** (A) The weekly percentage of participation in 2022, clearly indicating the first week of the FWC22 as the lowest point of participation for this year. (B and C) The distribution of weekly percentages of participation for the entire sCRC program from 2014 to 2022. The significant reductions in weeks 44 and 45 correspond to the first 2 weeks of the FWC22 when the Danish national team participated in comparison with the data for all years. (D) The difference in percentage of participation between women and men throughout 2022, with a clear decrease during the first 2 weeks of the FWC22 when the Danish national team participated.

believed, the Danish team could reach the semi-finals and maybe even the final. Many had organized gatherings with friends, alcohol, and partying. Wretchedly, the final outcome could not be further from expectations and Denmark finished last in its group during the first round.

There are many sporting events in Denmark with great public interest and media attention. Thus, the question here is why this event may have produced an effect that could be translated into making citizens forget, not attend to, or simply delay other activities? To our knowledge and after a thorough literature review, no previous studies have considered the potential association that an international sport event, such as FWC22 could have in terms of the level of negligence in attendance or participation reduction for health screening programs.

As previously mentioned, handball for both women and men is also a very popular sport in Denmark, in particular as the men's national team has won the World championship in 2019, 2021 and 2023. However, no reduction was detected during the weeks that these events took place, always in the month of January. Nevertheless, when comparing the time frames for the handball European championships (2014–2022), we noticed a decline during the first weeks of 2018, which partially coincided with a tournament during the same period (Fig. S1). However, similar reduction in participation has historically been observed during the last and first weeks of the year associated with the Christmas and New Years holidays. Therefore, this particular handball event association in the beginning of 2018 was ruled out, also as Denmark did not finish on the podium of finalists.

In the week, where Denmark had the first matches, only 34.2% of the recipients of invitations to sCRC have up till today returned a sample (Fig. 2A). To our surprise, the decline in participation in the sCRC was slightly more pronounced for women than for men. However, in overall, women had a higher participation in the screening program than men all year round (Fig. 3D). Roughly, 32% of the men and 38% of the women who received the invitation in the mailbox during the first week the Danish team played participated in sCRC. For the rest of the year percentages of participation were about 53.6% for men and 63% for women. Overall, the first 2 weeks of the FWC22 had significantly lower proportions of returned samples compared to the same weeks in all previous years since the screening program was initiated (Figs. 3B and 3C).

In July 2022, the Tour de France in that year edition began in Denmark with three stages, and the race even resulted with a Danish winner 23 days later. FWC22 and Tour de France probably have similar numbers of followers in Denmark; however, Tour de France did not prevent citizens from participating in the sCRC as no reduction was detected during the period of this event. In fact, during the period that Tour de France took place, the response time with citizens sending back their samples was among the lowest in 2022, meaning the citizens sent their stool samples quicker than the median time of 31 days.

We speculate that the reduction in participation for sCRC during FWC22 (compared with another major sport event, such as Tour de France) may be caused by potential differences in culture around the two events. For example, it is more common to meet up with friends and have a beer while watching a football match, than it is to do so while watching Tour de France, and possibly this social aspect pushes thoughts of returning samples further away from the minds of citizens. We encourage professionals from multidisciplinary fields across translational medicine such as social, biomedical, sports and natural science to combine efforts to help elucidate the unknown consequences of the potential intersection between football distracted enthusiasm and attendance in cancer screening programs.

Colorectal cancer screening programs worldwide are beneficial and improve the early detection of tumours and reduce mortality (Senore et al., 2019; Breekveldt et al., 2022; Bretthauer et al., 2022; Jauernik et al., 2023), and while there is some criticism about the economic cost of the different programs (Johansson, 2023), and the time they are require to run before reflecting an effective reduction in the incident and mortality (Bretthauer et al.,

*2022*; *Dominitz & Robertson, 2022*), the lives of the citizens saved by these initiatives must not be measured only in currency. Therefore, expanding knowledge about external factors that may influence and reduce participation in these programs is most relevant. While sports events are documented to provide benefits to health, it is surprising that little information can be found on the association that fans' enthusiasm can have on sCRC worldwide. The intersection of these two seemly random events, distracted football fans and a national sCRC, may have great implications for the medical translation into determining, understanding, and preventing external factors that affect the early detection of colorectal cancer and other human diseases. This may be the first study intersecting these two independent events, which is relevant as it may have potential consequences for early cancer detection.

## ACKNOWLEDGEMENTS

Thank you to all the staff at the Secretariat for sCRC at Region Zealand, Denmark for their extraordinary work. Special thanks to Berit Andersen, Michael Moran, Amish Acharya, Miguel Angel Garcia-Bereguiain and two anonymous reviewers for their feedback and comments on earlier versions of this manuscript.

### Funding

The authors received no funding for this work.

### Competing Interests

The authors declare they had no support from any organization for this work and no financial relationships with any organizations that might have an interest in this work in the previous 10 years. Additionally, authors declare that they are not involved with football at the international, national, local, or youth levels, nor other relationships or activities that could appear to have influenced this article.

### Author Contributions

- Alonzo Alfaro-Núñez conceived and designed the experiments, performed the experiments, analyzed the data, prepared figures and/or tables, authored or reviewed drafts of the article, and approved the final draft.
- Stina Christensen conceived and designed the experiments, performed the experiments, analyzed the data, authored or reviewed drafts of the article, and approved the final draft.
- Esther A. Jensen conceived and designed the experiments, performed the experiments, analyzed the data, authored or reviewed drafts of the article, and approved the final draft.

### Ethics

The following information was supplied relating to ethical approvals (*i.e.*, approving body and any reference numbers):

This research was approved by the Danish Data Protection Agency (J.No. 2023-52-0016).

## Data Availability

The data are available in the Supplemental File.

## Supplemental Information

Supplemental information for this article can be found online at http://dx.doi.org/10.7717/peerj.18057#supplemental-information.

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
