# Peer review of "Investigating the putative unforeseen link between football fervour and colorectal cancer screening in Denmark"

_PeerJ, doi:10.7717/peerj.18057_

## Round 0.1 · original submission · Minor Revisions

Three reviewers positively assessed the manuscript and basically I agree with their point. However, some beneficial comments were raised by them especially about methodology of data analysis. I hope the authors to answer these comments appropriately.

Reviewer 1 ·

Basic reporting

This manuscript was well-written in a clear and easily understandable English language.
The Introduction, while being short and concise, respects the audience's time by providing only the most relevant references for literature.
Methods are sufficiently described, and basic statistical analyses are used correctly.

For the Results section, I'd suggest adding more information.
Soccer fans are usually interested in other types of sports, and the Tour de France was mentioned, especially since the first three stages took place in Denmark in 2022.
To underline the statement that the decline in participation was due to the football World Cup, I'd suggest adding information about each football World Cup (2014 and 2018), European Championship (2016 and 2021), Winter Olympics (2014, 2018, and 2022), and Summer Olympics (2016, 2021) in the study period.
This could easily be achieved by placing icons in Supplemental Figure 1 like the football already inserted in the J panel. This would serve as a visual aid to prove your point. By the way, I can't find the reference for Supplemental Figure 1 in the main text? Please provide an appropriate reference to Supplemental Figure 1 in the main text.

The hypothesis is mentioned in the Discussion, but apparently, no other literature exists to confirm this; hence, the hypothesis remains purely speculative.

Experimental design

The article fits well with the aim and scope of PeerJ.
The research question is mentioned at the end of the introduction, but I suggest that no suggestions of causation are made, as there is no way of deciding the reasons for non-participation during the football World Cup.

Validity of the findings

The article has a novel take on the reason for non-participation in the colorectal cancer screening program. The conclusion is well stated, backing the notion that special care should be take to the attendance of cancer screening participation during a major sports event.

Reviewer 2 ·

Basic reporting

Basic reporting was ok.

Experimental design

I have some suggestions and comments there. It is provided as attachment.

Validity of the findings

I have some suggestions and comments there. It is provided as attachment.

Annotated reviews are not available for download in order to protect the identity of reviewers who chose to remain anonymous.

·

Basic reporting

Overall, this is a striking and interesting report with a great potential to influence public health policies. It is also very concise and well written.

Experimental design

The authors must be sure that there was no bias associated with the population invited for the screening during the world cup. As the authors have described, there are socio economical factors affecting those screenings like the education level (income level) of the population. I just want that the authors check if by any reasons, there could be a higher porcentaje of participants invited from lower income areas.

Validity of the findings

Please, referred to the comment in the section 2.

Additional comments

The authors could make some suggestions in the discussion to solve this public health issue. For instance, football players could be enroll in public health campaigns so we use the power of football in a good way to promote a healthy society.

---

## Round 0.2 · accepted · Accept

The authors responded the reviewers' comments appropriately.

Reviewer 1 ·

Basic reporting

No additional comments.

Experimental design

No additional comments.

Validity of the findings

No additional comments.

Additional comments

No additional comments.

Reviewer 2 ·

Basic reporting

My comments are well addressed.

Experimental design

My comments are well addressed.

Validity of the findings

My comments are well addressed.

·

Basic reporting

The authors have improved the manuscript following the reviewers' request.

Experimental design

The authors have improved the manuscript following the reviewers' request.

Validity of the findings

The authors have improved the manuscript following the reviewers' request.

Additional comments

The authors have improved the manuscript following the reviewers' request.